# Influence of Bone Morphology on In Vivo Tibio-Femoral Kinematics in Healthy Knees during Gait Activities

**DOI:** 10.3390/jcm11175082

**Published:** 2022-08-30

**Authors:** Sandro Hodel, Barbara Postolka, Andreas Flury, Pascal Schütz, William R. Taylor, Lazaros Vlachopoulos, Sandro F. Fucentese

**Affiliations:** 1Department of Orthopedics, Balgrist University Hospital, University of Zurich, Forchstrasse 340, 8008 Zurich, Switzerland; 2Laboratory for Movement Biomechanics, Institute for Biomechanics, ETH Zurich, 8093 Zurich, Switzerland

**Keywords:** knee, kinematic, pivot, bone morphology

## Abstract

An improved understanding of the relationships between bone morphology and in vivo tibio-femoral kinematics potentially enhances functional outcomes in patients with knee disorders. The aim of this study was to quantify the influence of femoral and tibial bony morphology on tibio-femoral kinematics throughout complete gait cycles in healthy subjects. Twenty-six volunteers underwent clinical examination, radiographic assessment, and dynamic video-fluoroscopy during level walking, downhill walking, and stair descent. Femoral computer-tomography (CT) measurements included medial condylar (MC) and lateral condylar (LC) width, MC and LC flexion circle, and lateral femoral condyle index (LFCI). Tibial CT measurements included both medial (MTP) and lateral tibial plateau (LTP) slopes, depths, lengths, and widths. The influence of bony morphology on tibial internal/external rotation and anteroposterior (AP)-translation of the lateral and medial compartments were analyzed in a multiple regression model. An increase in tibial internal/external rotation could be demonstrated with decreasing MC width β: −0.30 (95% CI: −0.58 to −0.03) (*p* = 0.03) during the loaded stance phase of level walking. An increased lateral AP-translation occurred with both a smaller LC flexion circle β: −0.16 (95% CI: −0.28 to −0.05) (*p* = 0.007) and a deeper MTP β: 0.90 (95% CI: 0.23 to 1.56) (*p* = 0.01) during the loaded stance phase of level walking. The identified relationship between in vivo tibio-femoral kinematics and bone morphology supports a customized approach and individual assessment of these factors in patients with knee disorders and potentially enhances functional outcomes in anterior cruciate ligament injuries and total knee arthroplasty.

## 1. Introduction

The native knee joint exhibits a range of motion in six degrees of freedom in the frontal, axial and sagittal planes [1]. Native knee joint kinematics are the result of a complex interaction of tibio-femoral bony anatomy and boundary conditions applied by the surrounding ligaments and muscles [1]. An understanding of the native knee motion characteristics is essential for the comprehension of pathologies and guiding surgical interventions around the knee joint.

The extent of tibial internal/external rotation during knee flexion and extension remains of special interest in anterior cruciate ligament (ACL) research. The increased anteroposterior (AP)-translation of the lateral compartment is the result of an asymmetrical anatomy with a more constrained, concave medial tibial plateau (MTP) and a relatively unconstrained, convex lateral tibial plateau (LTP) [2,3,4]. Various bone morphological factors have been investigated towards quantifying the anatomy of the femoral condyles, as well as tibial compartment concavity and convexity, and linked to native tibial internal/external rotation [4], ACL injury [5,6,7], or an increase in pivot-shift phenomenon [8,9]. Importantly, the axial rotation of the tibia relative to the femur is of high clinical relevance in the context of mimicking native motion characteristics following total knee arthroplasty (TKA) [10]. Here, aiming for natural kinematic behavior requires a detailed understanding of the physiological motion patterns and bone morphological variations of the healthy knee. Further insights into the relationships between bone morphology and native knee kinematics during daily activities under weight-bearing conditions could improve prevention and rehabilitation in ACL reconstruction. Identifying patients with increased axial rotation could reduce ACL failure and help refine the indications for additional anterolateral ligament reconstruction. With regards to TKA design, a more detailed personalized approach to restoring knee kinematics, especially in younger and active patients, can be expected.

Accurate assessment of native knee kinematics without the influence of soft tissue artifacts is challenging. Despite the development of novel accurate measurement techniques, only a limited number of studies have analyzed native knee kinematics with sub-degree and sub-millimeter accuracy during dynamic gait activities [11,12,13]. Different studies have reported contrary movement patterns, especially for the tibial internal/external rotation and the AP-translation of the medial and lateral compartments, as well as large inter-subject variability [11,14,15,16]. While individual limb alignment was not found as the predominant factor for governing individual tibial internal/external rotations or compartment translations, it is plausible that other anatomical variations of the femoral condyles and the tibial plateau are likely to serve as unidentified factors influencing native knee kinematics; however, the influence of bony morphology on native tibio-femoral kinematics during dynamic gait activities in healthy knees has not yet been investigated.

The primary aim of this study was, therefore, to quantify the influence of femoral and tibial bony morphology on tibial internal/external rotation and AP-translation of the lateral compartment during both the loaded stance phase and throughout full cycles of level walking in healthy knees using a video-fluoroscopic kinematic analysis. The rationale to define level walking as the primary outcome variable is based on the fact that most non-contact ACL injuries occur on level ground, while downhill activities result in increased eccentric hamstring activity [17] and potentially mask the influence of bony morphology. The secondary aim was to analyze the influence of the identified bony morphological factors on tibio-femoral kinematics during downhill walking, stair descent, and the unloaded swing phase of each of the three gait activities. Firstly, we hypothesized that bone morphology influences tibial internal/external rotation and AP-translation of the lateral compartment during the loaded stance phase and full gait cycles of level walking. Secondly, we hypothesized that identified bone morphological factors would also influence tibio-femoral kinematics during downhill walking, stair descent, and the swing phase of all three activities.

## 2. Materials and Methods

### 2.1. Subjects

All volunteers underwent clinical examination, video-fluoroscopic assessment, computer-tomography (CT), and long-legged radiograph of the lower limbs to allow the effect of bony morphology on tibio-femoral kinematics to be analyzed. The sample size was the result after a prospective enrolment for a previous tibio-femoral kinematic analysis [18]. A thorough clinical knee examination was performed by an experienced knee surgeon in advance to confirm the physiologically healthy range of motion (RoM) and ligament integrity. Inclusion criteria comprised healthy subjects without previous surgery or symptoms of the lower extremities and <9° varus–valgus malalignment. One subject did not complete all gait tasks and was excluded, leaving a total of 26 subjects (12 left, 14 right knees) for final analysis. The side included for analysis was chosen to allow a well-balanced distribution of varus/valgus alignment as defined for the previous study cohort [18]. Demographic characteristics, frontal, and axial leg alignment are summarized in Table 1.

### 2.2. Femoral and Tibial Bone Morphology

All subjects underwent CT of the analyzed knee (~20 cm proximal/distal of the joint line, resolution 0.5 × 0.5 mm, slice thickness 1 mm), including standardized reconstruction of three orthogonal planes (coronal, axial sagittal). To analyze the relationship between bone morphology and tibio-femoral kinematics, we included previously established radiological factors associated with native tibial internal/external rotation [4], ACL injury [5,6,7], or an increase in pivot-shift phenomenon [8,9]. The following bone morphological factors were measured from the subject-specific CT data, as described in the respective original publications (Figure 1):-*Femoral bone morphology:*

MC and LC width (in mm) [8] (Figure 1A) were measured in the coronal plane where both tibial spines are visible. MC and LC flexion circle diameter (in mm) [4] (Figure 1B,C) and lateral femoral condyle index (LFCI) [6] (Figure 1C) were measured on the mid-sagittal slice of each condyle (confirmed in the coronal plane).

-
*Tibial bone morphology:*


MTP and LTP width were measured (in mm) in the coronal plane, where both tibial spines are visible [8] (Figure 1A). MTP slope [7] (Figure 1B), depth [5] (Figure 1E), and length [8] (Figure 1E) were measured at the midsagittal slice of the MTP (confirmed in coronal plane) [5]. LTP slope [7] (Figure 1C), height [9] (Figure 1F) and length [8] (Figure 1F) were measured at the midsagittal slice of the LTP (confirmed in coronal plane). MTP and LTP slope were measured as the angle (°) formed by the articular surface of the MTP or LTP and a reference perpendicular to the midsagittal plane, defined by two circles tangent to the anterior, posterior tibial cortex, and the eminencia intercondylaris [7]. (Figure 1D) Medial to lateral slope asymmetry was calculated as described by Balcarek [19]. Positive values indicated a steeper slope of the MTP compared to the LTP.

All CT measurements were performed using a Picture Archiving and Communication System (PACS) (MERLIN version 5.8.1.1980, Phoenix PACS GmbH ©,Freiburg, Germany). Absolute measurements (mm) that correlated significantly with patient height (cm) (Pearson’s R) were scaled using the same factor as for scaling the AP-translation of tibio-femoral kinematic assessment, according to Gray et al. [11]. For inter-reader reliability, all measurements were performed by two independent readers blinded to the kinematic assessment.

Hip-knee-ankle angle, femoral antetorsion, and tibial torsion were measured using a biplanar X-ray system (EOS imaging, Paris, France) in a standing position [20].

### 2.3. Video-Fluoroscopy Set-Up and Motion Tasks

Motion tasks and tibio-femoral kinematics were analyzed using a moving fluoroscope for a previous study by Postolka et al. [18] and summarized hereafter. Radiographic images were acquired using a moving fluoroscope (25/30 Hz, 1 ms shutter time, image resolution 1000 × 1000 pixels, Figure 2) synchronized with a 22-camera optical tracking system [21] throughout complete cycles of three distinct gait tasks: level walking, downhill walking (10° declined slope), and stair descent (three steps, each 18 cm in height). Each participant performed a minimum of five valid cycles of each gait task and the average of these five cycles was used for further analysis.

### 2.4. Tibio-Femoral Kinematic Analysis

Three-dimensional (3D) models of all knees were created from CT by manual segmentation using open-source software (MITK-GEM) [22]. Fluoroscopic image data were corrected for distortion [23] before semi-automatic 2D/3D registration of the CT onto the fluoroscopic image was performed [24]. This process allows 3D reconstruction of the tibio-femoral kinematics with reported mean absolute errors of <1° for rotation and <0.6 mm for in-plane translations and <7.1 mm for out-of-plane translations. A femoral and tibial coordinate system was created according to Grood and Suntay [25] and identically aligned as previously described by Postolka et al. [13].

The condylar AP-position was computed for a medial (FFA_P_med_) and lateral (FFA_P_lat_) point on the functional flexion axis (FFA) of the femur. The functional flexion axis (FFA) of the femur was assessed using three deep knee bending trials and defined according to Ehrig et al. [26]. Relative AP-translation of the medial and lateral compartments was calculated as the movement of the FFA_P_med_ and FFA_P_lat_ with respect to the mid-coronal plane of the tibia. Tibial internal/external rotation (°) and AP-translation (mm) were then calculated during the loaded stance phase, the unloaded swing phase, and throughout the full gait cycle. The axial center of rotation (CoR) was calculated [27] and its mediolateral location was reported with respect to the tibial origin (−lateral/+medial (mm)). AP-translations were scaled to correct for each subject’s height by multiplication with a coefficient obtained by dividing the mean condylar width (all subjects) by the subject’s own condylar width [11].

### 2.5. Statistical Analysis

Numeric values are presented as mean ± standard deviation (SD) and categorical values as counts and percentages. Normal distribution was assessed using Shapiro–Wilk’s test. Inter-reader reliability was calculated using intraclass correlation coefficients (ICC) assuming a two-way mixed-effect and absolute agreement and graded according to Fleiss et al. [28] (>0.75 indicating excellent reliability). The mean values of both readers were used for further analysis. Correlation of absolute CT measurements (mm) with patient height (cm) was performed using Pearson’s correlation coefficient. To analyze the relationship between CT bony morphology (Figure 2), frontal, axial leg alignment, demographics (Table 1) and tibio-femoral kinematics, a stepwise linear regression was performed. To test our primary hypothesis, we defined the average tibial internal/external rotation and AP-translation of the lateral compartment of all subjects during full gait cycles and the loaded stance phase of level walking as the primary dependent variables. Identified significant predictors were then further included in regression models to analyze their effect on downhill walking, stair descent, and the unloaded swing phase solely. The absence of multicollinearity was tested by the variance inflation factor (VIF). Multicollinearity was defined if VIF ≥ 10. Fit of the regression model was reported as R^2^ and regression coefficients β (95% confidence interval (CI)) were reported. The significance was set at <0.05. Data were analyzed using SPSS version 26 (SPSS Inc., Chicago, IL, USA).

## 3. Results

### 3.1. Tibio-Femoral Kinematics

Mean tibial rotation changed from −9.0 ± 5.2° internal to 2.9 ± 5.0° external rotation during the loaded stance phase and from −9.4 ± 5.1° internal to 4.9 ± 5.3° external rotation during the complete cycle of level walking with a mean lateral AP-translation of 7.5 ± 1.9 mm (stance) and 10.0 ± 2.0 mm (complete cycle). The mean axial CoR throughout complete cycles of level walking was located 11.9 ± 6.3 mm medially to the origin of the tibial coordinate system.

### 3.2. Influence of Bone Morphology, Frontal, Axial Limb Alignment, and Demographics on Tibio-femoral Kinematics

The mean ICC for all CT measurements was 0.91 (range 0.77–1.00), indicating excellent reliability for all measurements.

An increase in tibial internal/external rotation could be demonstrated with a decreasing MC width β: −0.30 (95% CI: −0.58 to −0.03) (*p* = 0.03) during the loaded stance phase of level walking. An increased lateral AP-translation occurred with increasing medial to lateral slope asymmetry β: 0.50 (95% CI: 0.13 to 0.87) (*p* = 0.01) during the full gait cycle of level walking, and with a smaller LC flexion circle β: −0.16 (95% CI: −0.28 to −0.05) (*p* = 0.0007) and a deeper MTP β: 0.90 (95% CI: 0.23 to 1.56) (*p* = 0.01) during the loaded stance phase of level walking (Table 2). All factors analyzed during level walking are listed in the supplementary material. Frontal, axial leg alignment, age, gender, height, and weight did not significantly influence tibio-femoral kinematics.

The relationship of significant bony morphological factors with tibial internal/external rotation and lateral AP-translation during the loaded stance phase are visualized in Figure 3.

Subsequent analysis of the factors that were found to influence tibial internal/external rotation and lateral AP-translation during the loaded stance phase of level walking (Table 2) did not demonstrate a significant correlation with tibio-femoral kinematics during downhill walking, stair descent, or the unloaded swing phase solely.

## 4. Discussion

This is the first study investigating the link between femoral and tibial bone morphology and in vivo tibio-femoral kinematics throughout complete cycles of gait activities. The most important finding of our study is that a smaller LC flexion circle and a deeper MTP increased lateral AP-translation during the loaded stance phase of level walking. Additionally, subjects with a narrower MC demonstrated an increased tibial internal/external rotation during the loaded stance phase of level walking.

We identified a predominantly central to medially located axial CoR and a larger AP-translation of the lateral over medial compartment, which is in line with previous studies [4,11,29]. The described femoral and tibial bony morphological factors might explain previously described heterogeneous findings of tibial internal/external rotation and AP-translation of the lateral compartment between individuals and among studies [11,13,14]. LC flexion circle and MC width explained almost 40% of the reported variance of lateral AP-translation during the loaded stance phase. The influence of bony morphology on knee kinematics supports a customized approach and individual assessment of these factors in patients with knee disorders. This is especially of clinical relevance for patients with ACL injury as well as TKA candidates, as the axial tibial internal/external rotation has been identified as a crucial factor for non-contact ACL injury risk [30] and TKA design [31].

In accordance with previous studies, the identified bony morphological factors were associated with an increased ACL injury risk [5,6]. An increased tibial internal/external rotation, which can be assessed by the pivot-shift test, is a potential contributing factor for non-contact ACL injury risk or ACL re-rupture due to increased ACL ligament peak forces [32] and residual rotatory laxity after ACL-reconstruction [33]. This is of high clinical relevance for ACL risk screening, injury prevention, and might aid in identifying patients who benefit from concomitant anterolateral ligament reconstruction to restrain excessive rotational laxity [34,35,36]. In our cohort, a slope asymmetry with a steeper medial than lateral tibial slope increased lateral AP-translation during the full gait cycle but could not be reproduced during the loaded stance phase solely. The identified bony morphological factors did not play a significant role in tibio-femoral kinematics during downhill walking or stair descent in this cohort. This might be due to eccentric hamstring activation patterns for these activities [17]. Moreover, the analyzed factors did not influence kinematics during the unloaded swing phase, which underlines that bony geometry predominantly influences knee kinematics under load. Finally, frontal and axial limb alignment did not influence tibial internal/external rotation in this cohort, which might be due to the exclusion of patients with excessive malalignment or symptomatic mal-torsion.

Interpreting our findings in the context of TKA, considerable inter-individual differences in joint kinematics could be demonstrated that support a personalized approach. TKA design has evolved towards medially constrained designs, allowing a native femoral rollback of the lateral compartment during knee flexion [31,37] and possible preservation of the surrounding soft tissues [38]. The more natural kinematic behavior of these designs potentially helps to overcome unsatisfactory results following TKA [31]. In addition, individual implant selection and implant positioning could prove to be important factors for enhancing functional outcomes after TKA. As it remains unclear who will benefit from these designs to date, the described bony morphological factors add further evidence in the identification of appropriate candidates and optimizing patient selection in the future. In this study, the dynamic assessment using a moving video-fluoroscope allowed a highly accurate tibio-femoral kinematic analysis throughout full gait activities. The links between femoral and tibial bone morphology, and tibio-femoral kinematics, provide a unique insight into the relationships between anatomy and function during daily activities in healthy knees of a cohort that is well-balanced regarding frontal, axial leg alignment, and gender and therefore is representative of a normal population.

Several limitations should be considered when interpreting our findings. The complex set-up of a comprehensive radiographic and kinematic analysis based on a previous study resulted in a predefined, limited sample size; therefore, the reported findings need to be interpreted as univariate correlations rather than independent risk factors despite the use of a stepwise linear regression approach. All bone morphology parameters were measured in CT, which might have influenced our findings as the original measurements were described in magnetic-resonance imaging. Nevertheless, all CT measurements indicated excellent reliability. Regarding the kinematic assessment, variable definitions of coordinate systems and the use of different kinematic approaches limit direct comparison of our findings to existing literature to some extent. Potential sources of error are the limited out-of-plane accuracy induced by the single-plane video-fluoroscopic set-up. Although this plausibly affects the determination of the mediolateral CoR, relative rotations and translations are less influenced. Nevertheless, kinematic assessment using moving fluoroscopy represents the state-of-the-art in understanding joint motion patterns over complete cycles of gait activities to date.

## 5. Conclusions

Evidence presented in this study suggests that specific characteristics of femoral and tibial bone morphology influence tibio-femoral kinematics in healthy knees. A narrower MC width increases tibial internal/external rotation during the loaded stance phase of level walking. A smaller LC flexion circle and a deeper MTP increase lateral AP-translation of the knee during the loaded stance phase of level walking. The identified relationship between in vivo tibio-femoral kinematics and bone morphology in healthy knees supports a customized approach and individual assessment of these factors in patients with knee disorders and potentially enhances functional outcomes in anterior cruciate ligament injuries and total knee arthroplasty.

## Figures and Tables

**Figure 1 jcm-11-05082-f001:**
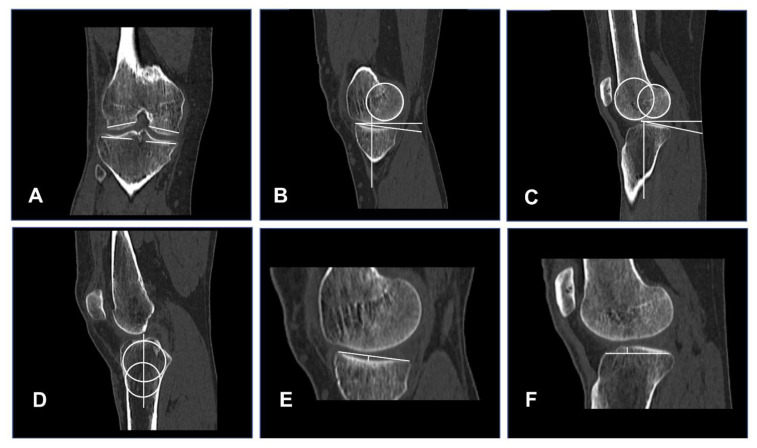
Measurement of femoral and tibial bony morphological factors in computer tomography. Femoral and tibial bony morphology measurements according to the original publication and as described in the text: (**A**): Medial and lateral femoral condylar width and tibial plateau width (mm) [8]. (**B**): Medial femoral flexion circle diameter (mm) and medial tibial plateau slope (°) [4,7]^.^ (**C**): Lateral femoral condyle index, lateral flexion circle diameter (mm) [4,6] and lateral tibial plateau slope (°) [7]. (**D**): Midsagittal tibial reference for the medial and lateral tibial plateau slope measurement [7]. (**E**): Medial tibial plateau length [8] and depth (mm) [5]. (**F**): Lateral tibial plateau length [8] and height (mm) [9].

**Figure 2 jcm-11-05082-f002:**
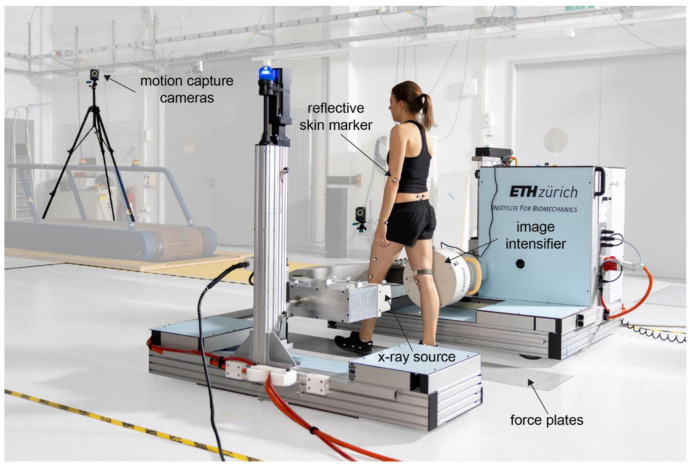
Video-fluoroscopy set-up during level walking (reprinted from Postolka et al. [13]). Tibio-femoral kinematics were assessed using a moving fluoroscope set-up synchronized with a 22-camera optical tracking system. Each subject completed three gait tasks: level walking, downhill walking, and stair descent.

**Figure 3 jcm-11-05082-f003:**
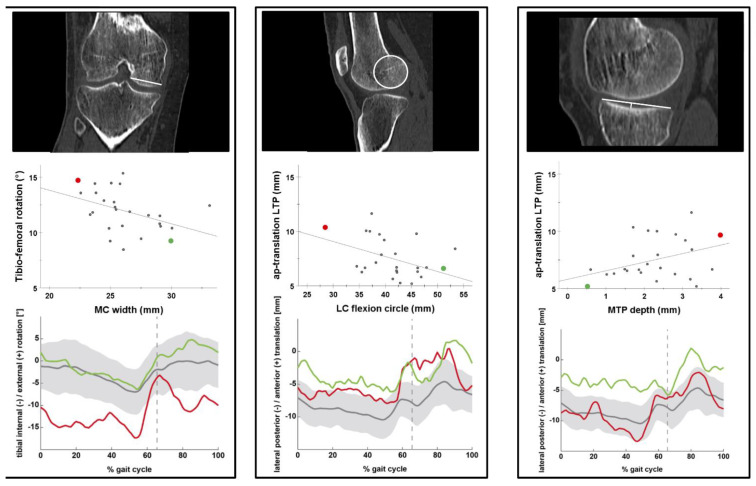
Relationship between tibio-femoral kinematics and medial condylar width, lateral condyle flexion circle, and medial tibial depth during the loaded stance phase of level walking. **Left box**: Medial condyle (MC) width. **Middle box**: Lateral condyle (LC) flexion circle. **Right box**: Medial tibial plateau (MTP) depth. Top row visualizes bony morphology, middle row depicts scatterplot, and bottom row tibial internal/external rotation or lateral AP-translation during full gait cycles of level walking (thick grey line: mean, grey shaded: ±SD, dashed vertical line: mean toe-off). Red dot/line: average of individuals demonstrating the highest tibial internal/external rotation or lateral anteroposterior (AP)-translation during loaded stance phase. Green dot/line: average of individuals demonstrating the lowest tibial internal/external rotation or lateral AP-translation during the loaded stance phase.

**Table 1 jcm-11-05082-t001:** Demographic characteristics of the study participants.

	26 Subjects (26 Knees)
Age (years)	27.3 ± 10.5 (20 to 67)
Male	13
Female	13
BMI (kg/m^2^)	21.3 ± 2.2 (17 to 26)
Height (cm)	177.6 ± 9.3 (162.0 to 195.2)
Weight (kg)	67.3 ± 11.0 (50.8 to 93.6)
HKA (°)	+1.4 ± 4.1 (−8.0 to +9.0)
Femoral antetorsion (°)	17.8 ± 11.4 (−6 to +52)
Tibial torsion (°)	31.3 ± 7.4 (13 to 44)

HKA: Hip-knee-ankle angle (+ varus/− valgus). Femoral antetorsion (+ antetorsion/− retrotorsion). BMI: Body mass index. Data reported mean ± SD (range) or counts if not stated otherwise.

**Table 2 jcm-11-05082-t002:** Bony morphological factors associated with tibial internal/external rotation and lateral AP-translation during full gait cycle and loaded stance phase of level walking.

Kinematic Outcome Variable	Regression Model Fit (R^2^; *p*-Value)	Factors Included in Regression Model β (95% CI)	*p*-Value (β)
**Tibial internal/external rotation (°)**			
during full gait cycle	NA	NA	NA
during loaded stance phase	0.18; 0.03	**MC width:** −0.30 (−0.58 to −0.03)	**0.03**
**Lateral AP-translation (mm)**			
during full gait cycle	0.24; 0.01	**Tibial slope asymmetry:** 0.50 (0.13 to 0.87)	**0.01**
during loaded stance phase	0.38; 0.004	**LC flexion circle:** −0.16 (−0.28 to −0.05)	**0.007**
		**MTP depth:** 0.90 (0.23 to 1.56)	**0.01**

AP: Anteroposterior. NA: Not applicable. MC: Medial condyle. LC: Lateral condyle. MTP: Medial tibial plateau. Significant factors included in the regression model marked **bold** (*p* < 0.05).

## Data Availability

The data presented in this study are available on request from the corresponding author.

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
