# Peer review of "Influence of Bone Morphology on In Vivo Tibio-Femoral Kinematics in Healthy Knees during Gait Activities"

_jcm, 2022, doi:10.3390/jcm11175082_

Round 1
Reviewer 1 Report
Paper presents a study for evaluating the relationship between in vivo tibio-femoral kinematics and bone morphology. Tests were performed on a cohort of 26 healthy subjects. Paper is very well-structured and written, as well as the topic is interesting and with an high level of innovation.
Authors are just invited to deeply revise the Introduction in order to better contestualize the topic of the paper and to let readers understanding the significance of the results and the importance of the study. Consequently, also the Discussion can be enriched by tangible outcomes and applications.
Reviewer 2 Report
Hodel et al. studied the relation between CT measured femoral and tibial bone morphology in relation to in-vivo kinematics during gait activities. A narrower MC width increases tibial rotation during the loading stance phase, and a smaller LC flexion circle and deeper MTP during the walking phase. The aim is clear, however, the authors should take overfitting their models with too much predictors into account, as the limited sample size may introduce bias.
Major: It is not correct to state that the relations in the current study are independent relations (irrespective of patient characteristics). With a sample size of n=26, conclusions can only be drawn from univariate associations.
Minor:
- Please include the exact number of patients that were excluded for each catergory (previous surgery, symptoms, varus/valgus malalignment)
- It is unclear which side of knees were included, and if all patients had the same side analyzed. Please specify.
Round 2
Reviewer 2 Report
I would like to thank the authors for revising their manuscript accordingly.
Author Response
Dear all,
Thank you very much for the review. The only point is minor spell check, as I noticed. I would like to point out that the manuscript has been thoroughly reviewed by William R Taylor, one of the contributing senior authors, who is a native English speaker. If any specific concerns regarding style or language persist, we are happy to adress those.
Sincerely,
Sandro Hodel